# The genetic basis of the kākāpō structural color polymorphism suggests balancing selection by an extinct apex predator

Lara Urban[1,2,3,4]*, Anna W. Santure[5], Lydia Uddstrom[6], Andrew Digby[6], Deidre Vercoe[6], Daryl Eason[6], Jodie Crane[6], Kākāpō Recovery Team[¶], Matthew J. Wylie[7,8], Tāne Davis[6,7], Marissa F. LeLec[9], Joseph Guhlin[9], Simon Poulton[10], Jon Slate[11], Alana Alexander[4], Patricia Fuentes-Cross[4], Peter K. Dearden[9], Neil J. Gemmell[4], Farhan Azeem[12,13], Marvin Weyland[12,13], Harald G. L. Schwefel[12,13], Cock van Oosterhout[14☽], Hernán E. Morales[15,16☽]

1 Helmholtz AI, Helmholtz Munich, Neuherberg, Germany, 2 Helmholtz Pioneer Campus, Helmholtz Munich, Neuherberg, Germany, 3 Technical University of Munich, School of Life Sciences, Freising, Germany, 4 Department of Anatomy, University of Otago, Dunedin, New Zealand, 5 School of Biological Sciences, University of Auckland, Auckland, New Zealand, 6 Kākāpō Recovery Programme, Department of Conservation, Invercargill, Murihiku, Aotearoa New Zealand, 7 Ngāi Tahu, Ngāti Māmoe, Waitaha, New Zealand, 8 The New Zealand Institute for Plant and Food Research Limited, Nelson, New Zealand, 9 Genomics Aotearoa and Department of Biochemistry, University of Otago, Dunedin, New Zealand, 10 School of Biological Sciences, University of East Anglia, Norwich, United Kingdom, 11 School of Biosciences, University of Sheffield, Sheffield, United Kingdom, 12 Department of Physics, University of Otago, Dunedin, New Zealand, 13 Dodd-Walls Centre for Photonic and Quantum Technologies, Dunedin, New Zealand, 14 School of Environmental Sciences, University of East Anglia, Norwich Research Park, Norwich, United Kingdom, 15 Globe Institute, Faculty of Health and Medical Sciences, University of Copenhagen, Copenhagen, Denmark, 16 Department of Biology, Ecology Building, Lund University, Lund, Sweden

☽ These authors contributed equally to this work.
¶ Membership of Kākāpō Recovery Team is provided in the Acknowledgments.
* lara.h.urban@gmail.com

**Data Availability Statement:** The kākāpō population genomic dataset and genetic variant callset are available via an application form: https://www.doc.govt.nz/our-work/kakapo-recovery/what-

## Abstract

The information contained in population genomic data can tell us much about the past ecology and evolution of species. We leveraged detailed phenotypic and genomic data of nearly all living kākāpō to understand the evolution of its feather color polymorphism. The kākāpō is an endangered and culturally significant parrot endemic to Aotearoa New Zealand, and the green and olive feather colorations are present at similar frequencies in the population. The presence of such a neatly balanced color polymorphism is remarkable because the entire population currently numbers less than 250 birds, which means it has been exposed to severe genetic drift. We dissected the color phenotype, demonstrating that the two colors differ in their light reflectance patterns due to differential feather structure. We used quantitative genomics methods to identify two genetic variants whose epistatic interaction can fully explain the species' color phenotype. Our genomic forward simulations show that balancing selection might have been pivotal to establish the polymorphism in the ancestrally large population, and to maintain it during population declines that involved a severe bottleneck. We hypothesize that an extinct apex predator was the likely agent of balancing selection, making the color polymorphism in the kākāpō a "ghost of selection past."

we-do/research-for-the-future/kakapo125-gene-sequencing/request-kakapo125-data/. Access to the data is controlled by a data committee composed of the Aotearoa New Zealand Department of Conservation and Te Rūnanga o Ngāi Tahu. The computational code (including Python v3.5.2 code) is available via Github: https://github.com/LaraUrban42/Kakapo_genomics.

**Funding:** The Alexander von Humboldt Foundation (grant number DEU 1209620 FLF-P) provided financial support of LU for study design, data collection and analysis, and preparation of the manuscript. The European Research Council (ERODE, grant number 101078303) provided financial support of HEM for data analysis and preparation of the manuscript. Genomics Aotearoa (https://www.genomics-aotearoa.org.nz/) provided financial support for the project including for AWS, JG, PKD and NJG for study design, data collection and analysis, and preparation of the manuscript. The funders did not play any role in the study design, data collection and analysis, decision to publish, or preparation of the manuscript. Views and opinions expressed are those of the authors only and do not necessarily reflect those of the European Union or the European Research Council; neither the European Union nor the granting authority can be held responsible for them.

**Competing interests:** The authors have declared that no competing interests exist.

**Abbreviations:** API, application programming interface; AU-ROC, area under the receiver operating characteristic curve; BP, balanced polymorphism; ENA, European Nucleotide Archive; GWAS, genome-wide association study; HWE, Hardy–Weinberg equilibrium; LD, linkage disequilibrium; MAF, minor allele frequency; MCMC, Markov chain Monte Carlo; NFDS, negative frequency-dependent selection; PCA, principal component analysis; SEM, scanning electron microscopy; SNP, single-nucleotide polymorphism.

## Introduction

About a century ago, population geneticists began to study the evolution of polymorphisms at genetic loci [1]. We have now reached a milestone where we can examine nucleotide variation across the entire genome of every single individual of a species. This remarkable feat has been accomplished for the kākāpō (*Strigops habroptilus*) [2], a critically endangered flightless parrot species endemic to Aotearoa New Zealand that is considered a taonga (treasure) of Ngāi Tahu, a Māori iwi (tribe) of Te Wai Pounamu (the South Island of Aotearoa) [3–5]. The genomic data that has been generated for this species now enables us to study population genomics at an unprecedented resolution, improving our understanding of the evolutionary forces impacting this threatened species with the potential to inform their conservation and recovery in partnership with Ngāi Tahu [2,4,6,7].

In this study, we explore the puzzling phenomenon of the kākāpō feather color polymorphism ("green" and "olive" phenotypes) and how this severely bottlenecked species managed to retain its phenotypic diversity. The kākāpō has experienced a population decline and significant genetic drift over the past 30,000 years (30 KYA), followed by a sharp, anthropogenically induced population bottleneck resulting in just 51 surviving individuals in 1995 [7]. Under the close management of the Aotearoa New Zealand Department of Conservation Te Papa Atawhai Kākāpō Recovery Programme in partnership with Ngāi Tahu [4,8], the population size has increased to 247 birds located on several predator-free islands [as of August 1, 2024]. Despite these severe bottleneck events, the population exhibits a relatively balanced distribution of green and olive individuals; to understand the evolution of this color polymorphism, we here assess the phenotype's genomic basis and how the genetic polymorphism was first established and has since been maintained in the population.

Population genetic theory hereby stipulates that newly emerged genetic variants (i.e., mutations) are rarely expected to reach appreciable allele frequencies when they evolve neutrally since they often get lost through genetic drift [9]. Kimura's neutral theory of molecular evolution suggests that most genetic polymorphisms are neutral (or slightly deleterious) because if a mutation conferred a selective advantage, it would be expected to rise to fixation, replacing the ancestral allele [10]—unless the frequency rise of such beneficial variants is halted by one or more counteracting evolutionary forces. Balancing selection can theoretically balance allele frequencies and produce stable genetic polymorphisms, for example, through negative frequency-dependent selection (NFDS), overdominance, antagonistic selection, or temporally and spatially varying selection [11]. To understand the establishment and maintenance of genetic polymorphisms, we therefore need to first reject the null hypothesis of neutral evolution, and then describe the evolutionary forces that help maintain the genetic polymorphism [11–13]. In the case of the kākāpō's genetic polymorphism that underlies its color polymorphism, we therefore have to assess how the genetic polymorphism was not randomly lost in the species' large ancestral population, and how it has since been maintained despite population decline and a severe population bottleneck.

Understanding the establishment and maintenance of the kākāpō polymorphism might be of importance for ongoing conservation efforts since the significant majority (nearly 70%) of the wild founder population from 1995 were of green color. Whether the color polymorphism is likely to impact fitness in the absence of current intensive conservation management is therefore of potential conservation relevance—especially given the long-term plans of the Kākāpō Recovery Programme and Ngāi Tahu to restore this critically endangered species beyond predator-free islands. Why the kākāpō species has maintained the green and olive phenotype has so far remained elusive. Before the arrival of tūpuna Māori in AD 1280 [14], the only predators of the kākāpō were avian [15], the vast majority of which hunt by sight. While

the apex predators, the Haast's eagle (*Hieraaetus moorei*) and the Eyles' harrier (*Circus teauteensis*), went extinct approximately 600 years ago [14], we hypothesize that the kākāpō color polymorphism could be a consequence of selective pressures through this past predation. For example, NFDS through search image formation in the avian predators can theoretically maintain such polymorphisms [16].

To explore which hypothesis best explains the kākāpō color polymorphism, we created and analyzed detailed phenotypic data of kākāpō plumage together with high-coverage genomic data of nearly the entire kākāpō species (*n* = 169; as of January 1, 2018) [2,11]. We found that two epistatically interacting single-nucleotide polymorphisms (SNPs) at the end of chromosome 8 can explain the color polymorphism of all individuals. A candidate gene analysis in this genomic region allowed us to hypothesize a structural color polymorphism, which we explored with subsequent optical analyses. The nucleotide divergence between the two indicator haplotypes predicted that the color polymorphism evolved circa 1.93 MYA, around the time when the avian kākāpō predators evolved. Using genomic forward simulations, we determined that the polymorphism's establishment is highly unlikely under neutral evolution, while any selective advantage of the novel color morphology would have likely led to its rapid fixation. We show that the effects of balancing selection would have been sufficient to establish the polymorphism in the large ancestral population. Our simulations show that the polymorphism could have subsequently been maintained despite declining population size and a severe kākāpō population bottleneck. Also this polymorphism has been maintained despite dramatic ecological changes in Aotearoa New Zealand approximately 600 years ago when several indigenous bird species including the major natural kākāpō predator species became extinct [14]. Based on this evidence, we propose that now-extinct apex predators were the likely agent of balancing selection, which would make the color polymorphism in the kākāpō a "ghost of selection past."

## Results

### Quantitative genomics

**Phenotypic data.**   We firstly created a catalog of the feather color polymorphism of the kākāpō species. Wherever possible, we did so through standardized photography of living birds followed by manual annotation (**Fig 1A** and Material and Methods); in the case of deceased birds, we used historical photographs (**S1 Table**, Photography sheet). As green and olive individuals are not always easy to distinguish, we used Commission on Illumination $L^*a^*b^*$ (CIELAB) color space analysis, which projects any color into a 2D color space $a^*b$ that is perpendicular to and therefore independent of luminance $L$ to define objective criteria of green and olive colors (Material and Methods). By analyzing photographs of kākāpō with distinct green or olive color, we found that the median and spread of $a$ is predictive of the color morphology (Material and Methods). We then leveraged this CIELAB analysis to annotate all kākāpō individuals for which we had any doubt in terms of their phenotypic classification (**S1 Table**, CIELAB sheet).

**Genomic data.**   Our previous population genomic analyses identified 2,102,449 SNPs in Illumina short-read sequencing data across 169 kākāpō individuals using the reference genome bStrHab1.2.pri (NCBI RefSeq assembly accession number GCF_004027225.2) [2,5,7] and the genetic variant calling tool DeepVariant [2,17]. We subsequently excluded the kākāpō individual Konini 3-4-16 since this bird died at 26 days of age, i.e., before its feather color could be determined (kākāpō only possess white down when hatching and adult plumage color becomes apparent at about 50 days of age). We filtered the SNP set according to sequencing depth, quality, minor allele frequency (MAF), genotype missingness, biallelicity, and

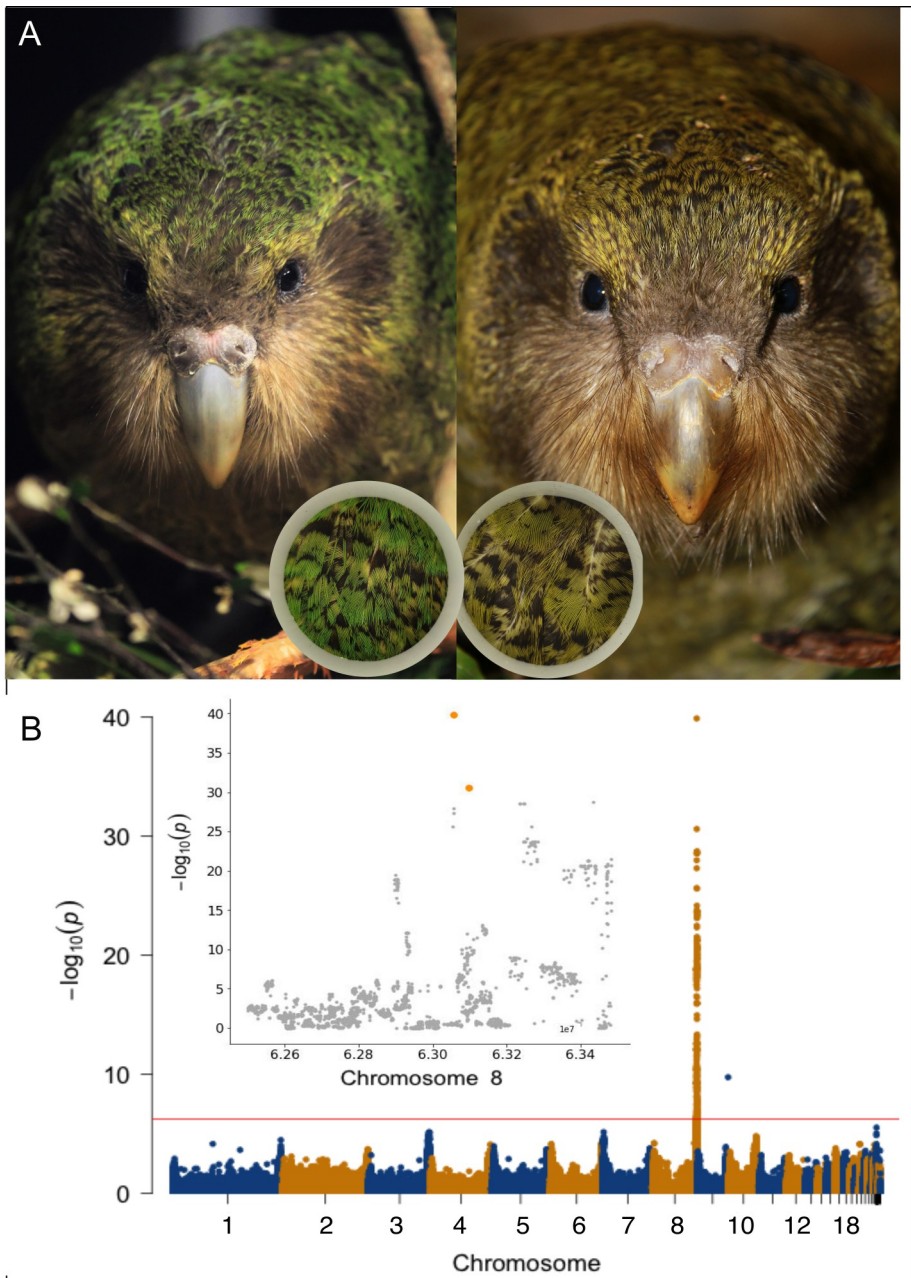

**Fig 1. The kākāpō feather color polymorphism.** (A) Green (left) and olive (right) kākāpō individuals; the photographs show the kākāpō individuals Uri (left) and Bravo (right) who are full siblings. The inserts show the standardized photography that was applied for color assessment across the extant kākāpō population (Material and Methods). (B) Manhattan plot of the mixed-model GWAS between all bi-allelic SNPs and the binary color polymorphism phenotype (Material and Methods); the horizontal line indicates the genome-wide significance line after Bonferroni multiple testing correction. The genome-wide significant hit at the end of chromosome 9 represents a single SNP, which we therefore discarded as noise. Inlet: Zoom-in of the Manhattan plot on the end of chromosome 8 (from $6.25 \times 10^{7}$ bp to the end of the chromosome); the two most significant SNPs Chr8_63055688 and Chr8_63098195 are highlighted in orange color. All photographs are originals taken by the authors of this manuscript. The code to generate this figure can be found in https://zenodo.org/records/13302801. For data see the "Data and code availability" section. GWAS, genome-wide association study; SNP, single-nucleotide polymorphism.

extreme deviations from Hardy–Weinberg equilibrium (HWE) to obtain a final set of 1,211,090 high-quality SNPs (Material and Methods). We visualized this genomic data through principal component analysis (PCA) [14], whose first 10 axes did not show any obvious genome-wide difference between green and olive individuals (**S1A Fig** and Material and Methods).

**Genome-phenome associations.** We used Bayesian multiple regression modeling as implemented in BayesR [18] to estimate the heritability of the kākāpō color polymorphism at 77.7% (95% credible interval of 59.9% to 89.2%; Material and Methods). The chromosome partitioning analysis, which calculates heritability per chromosome, shows that nearly the entire genetic variance (>99%) lies on chromosome 8 (**S1B Fig** and Material and Methods). Within-population genomic prediction of color using 10-fold cross-validation in BayesR resulted in an area under the receiver operating characteristic curve (AU-ROC) of 0.99 (**S1C Fig** and Material and Methods).

We then used RepeatABEL [19] to perform a mixed-model genome-wide association study (GWAS) across all 168 kākāpō individuals while accounting for relatedness and other random as well as fixed effects (Material and Methods). We identified one genomic region strongly associated with color polymorphism (**Fig 1B**) that contained the top-hit SNP Chr8_63055688 ($p = 10^{-40}$). As RepeatABEL is based on linear mixed-effect models which do not directly account for binary-dependent variables, we confirmed this result using PLINK [14], which only considers fixed effects, but directly models binary phenotypes through logistic regression; we did not identify any further genome-wide significant associations with a data set of small insertions and deletions (indels) generated by Guhlin and colleagues [2] (Material and Methods).

The top SNP for both the heritability test and the GWAS, Chr8_63055688 (A|T alleles), explains the color polymorphism of nearly the entire extant kākāpō population; only the phenotypes of 5 individuals (Egilsay, Elliott, Percy, Dusky, and Te Atapō) were misclassified when assuming a dominant effect of the alternative T allele encoding the olive color morphology (**S1 Table**, *GWAS* sheet). For example, Te Atapō is heterozygous but of green color. We therefore applied a genetic algorithm [20] to 1,100 SNPs located in or close (within 400 k nucleotides) to the significantly associated genomic region to find combinations of SNPs that can fully describe the phenotypic distribution (Material and Methods). This test identified a dominant-epistatic interaction with a second SNP as the most probable genomic architecture of the color polymorphism. This SNP, Chr8_63098195 (T|G alleles), was the second-best GWAS hit ($p = 10^{-31}$). While the phenotype of the kākāpō individuals Egilsay and Elliott could first not be explained correctly by the suggested dominant-epistatic interaction, we investigated our data further and identified a genomic sample swap between these 2 individuals. Taking this sample swap into account, the color polymorphism of every kākāpō individual can be explained by the genetic rule that olive individuals must carry at least 1 alternative T allele at Chr8_63055688 and at least 1 alternative G allele at Chr8_63098195 (**S1 Table**, *GWAS* sheet). These 2 indicator SNPs are in strong linkage disequilibrium (LD) ($r^2 = 0.696$) but are interspersed with genomic regions of low LD ($r^2 < 0.1$; Materials and methods and **S2A Fig**); therefore, the entire genomic region containing both SNPs is not under strong linkage (i.e., undergoes recombination). To discard the impact of any large genomic rearrangements such as inversions or of other uncalled genetic variants, we investigated the mean sequencing depth across individuals in this genomic region. The mean depth as captured through a sliding-window approach remains stable across the genomic region, with a few decreases in depth at individual genetic sites (**S2B Fig** and Material and Methods).

**Out-of-sample predictions.** We next validated the epistatic effect of the 2 indicator SNPs on the color morphology (Material and Methods), again assuming the genetic rule that olive

individuals must carry at least 1 alternative T allele at Chr8_63055688 and at least 1 alternative G allele at Chr8_63098195 (**S1 Table**, *GWAS* sheet). As Te Atapō was the only kākāpō whose green phenotype could be explained by reference homozygosity at Chr8_63098195 (TT) while heterozygosity at the top SNP Chr8_63055688 (AT) would have predicted an olive phenotype, we analyzed the genomic and phenotypic data of all offspring of Te Atapō to confirm our predicted epistatic effect: Te Atapō (genotypes for Chr8_63055688 and Chr8_63098195 and phenotype: AT, TT, green) had offspring with Pura (AT, TG, olive) and Tumeke (AA, TT, green). The offspring of Te Atapō and Pura were Uri (AT, TT, green; **Fig 1A**, left), Bravo (TT, TG, olive; **Fig 1A**, right), and Hanariki (TT, TG, olive); the offspring of Te Atapō and Tumeke were Tutū (AA, TT, green) and Meri (AA, TT, green). These observations support the hypothesis of the dominant-epistatic effect of the indicator SNPs on the color morphology in kākāpō.

We further investigated the genomic underpinnings of a deceased yellow kākāpō individual to examine its genotype (Material and Methods). We found this individual to be genetically olive (AT, TG), and the yellow color might therefore be a consequence of discoloration, which is known to occur in parrots due to malnutrition and viral diseases [21].

**Candidate gene analysis.** We next identified all genes in the annotated genomic regions to find underlying candidate genes (**S2 Table** and **S2C Fig**). Both indicator SNPs lie in predicted intergenic and intronic regions, respectively, so their molecular functional consequence is not straightforward to assess. Furthermore, the gene *TNNI3K*, in whose first intron the SNP Chr8_63098195 falls, has weak annotation support according to the NCBI XM track (NCBI RefSeq assembly accession number GCF_004027225.2) and the RNA-seq exon coverage (**S2C Fig**).

The most likely gene to be involved in color polymorphism according to its known function is *LHX8*, which lies approximately 200 k nucleotides downstream of the indicator SNPs. The protein encoded by this gene is a transcription factor and a member of the LIM homeobox family of proteins, which contains 2 tandemly repeated cysteine-rich double-zinc finger motifs known as LIM domains in addition to a DNA-binding homeodomain. *LHX8* is known to be involved in the patterning and differentiation of various tissue types and plays a role in tooth morphogenesis (**S2 Table**). This candidate gene therefore made us hypothesize that a structural color morphology might underlie the kākāpō color polymorphism, which we followed up on with optical analyses.

## Optical analyses

To evaluate the hypothesis of a structural color morphology in kākāpō feathers and its potential functional consequences, we used scanning electron microscopy (SEM) to investigate the surface of 3 olive and 3 green feathers (**Fig 2A** and Material and Methods). The smoothness analyses of the SEM pictures at 5,500× resolution (Material and Methods) resulted in mean gradient magnitude values across the 6 analyzed feathers of 29.00 for green (individual values from left to right; **Fig 2A**: 37.28; 24.31; 25.40) and of 43.01 for olive (individual values from left to right; **Fig 2A**: 49.98, 30.99, 48.07) feathers (exact one-sided Mann–Whitney U test, $p = 0.1$; Material and Methods). While we cannot establish statistical significance at $\alpha = 0.05$ based on the 6 available data points, this analysis points to a reduced average gradient magnitude, i.e., increased smoothness, of the green feathers.

We then subjected the feather tips of 1 green and 1 olive feather to near-infrared/visible photoreflectometry to assess differences in light reflectance (**Fig 2B**); we indeed found evidence that besides differences in the visible wavelength spectrum (400 to 700 nm), the olive feather reflected less light in the UV spectrum (<400 nm; **Fig 2B**). Taken together with the known difference in light reflectance in the visible spectrum (see Results/Phenotypic data),

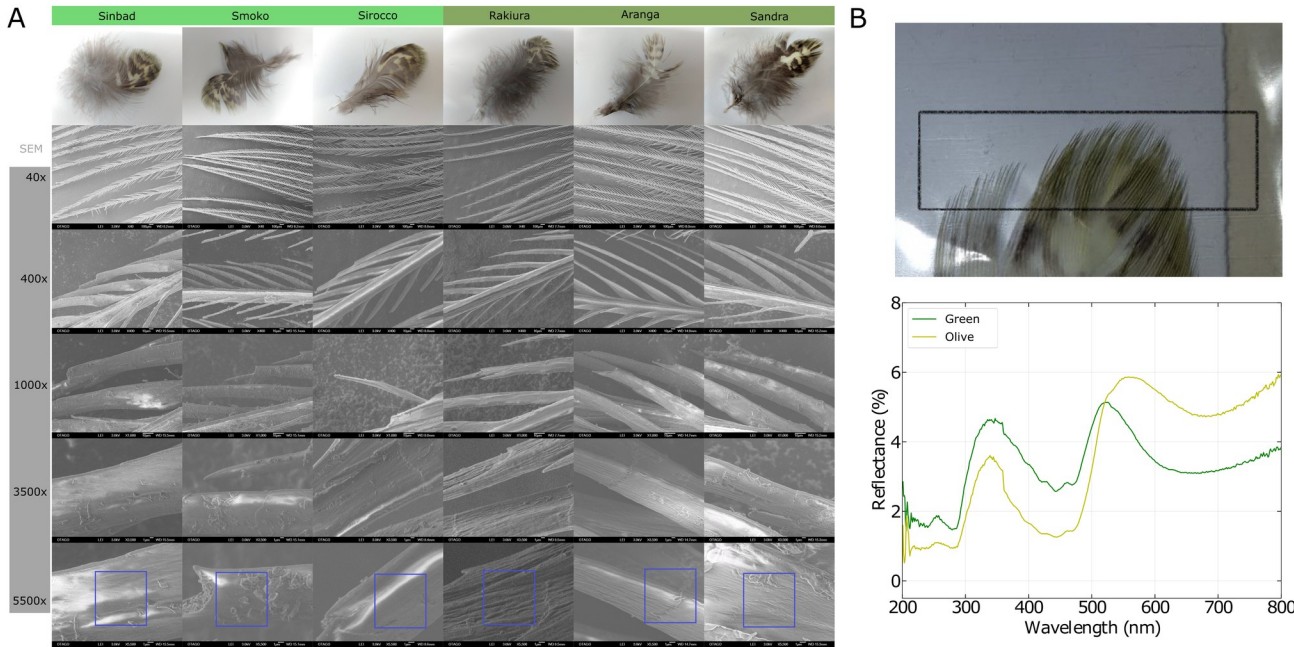

**Fig 2. Optical analyses of kākāpō feathers of both color polymorphisms.** (A) SEM images of 3 green (left 3 columns) and 3 olive (right 3 columns) feather barbules at different resolutions (rows); the green feathers show a smoother surface than the olive ones. (B) Photoreflectometry of near-infrared/ visible wavelengths in the feather tips; the relative reflectance of an exemplary olive and green is plotted over the wavelength of the reflected light (Material and Methods). All photographs are originals taken by the authors of this manuscript. The data underlying this figure can be found in https://zenodo.org/records/13302801. SEM, scanning electron microscopy.

this points to potential differential fitness consequences between the color morphologies from differential light reflectance patterns—also given that many avian and other predator species can see UV besides visible light [22].

## Color polymorphism evolution

**Evolutionary origin.** We identified the green allele to be the ancestral allele of our indicator SNPs of interest by leveraging the high-quality reference genome of the most closely related species, the kea (*Nestor notabilis*), which diverged from the kākāpō about 60 to 80 MYA [23] (Material and Methods). Using principles of evolutionary genetic theory about haplotype divergence that allow to estimate divergence times based on a molecular clock [10] and assuming a generation time of 15 years, we estimated the genetic polymorphism to have arisen around 128,500 generations ago (95% confidence interval of 39,300 to 302,000 generations) or 1.93 MYA (0.59 to 4.52 MYA) (Material and Methods).

**Establishment and maintenance of color polymorphism.** We tested competing neutral and adaptive hypotheses for the establishment and maintenance of the color polymorphism with individual-based forward simulations in SLiM3 [24] (Material and Methods). We found that the probability of establishment (i.e., not lost by genetic drift) of the color polymorphism in a large ancestral population ($N_e = 36,000$ [7]) was less than one in a million if we assume neutral evolution (**S3A Fig**). If we assumed a selective advantage, the probability increased modestly; for a small selective advantage of 1%, the novel olive color had an establishment probability of approximately 500 in a million. For a much larger selective advantage of 20%, the probability increased to nearly approximately 5,000 in a million. While the probability of establishment is therefore overall low and requires a selective advantage (**S3A and S3B Fig**),

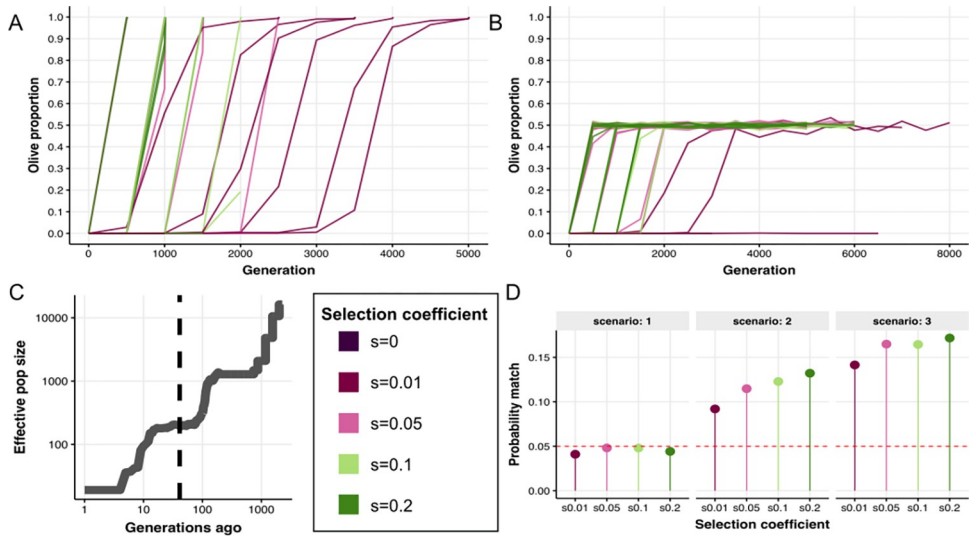

**Fig 3. Genomic forward simulations of color polymorphism evolution.** The simulations assume (A) a positive selection model, where the novel olive phenotype had a selective advantage ranging between 1% and 20%, and (B) a balancing selection model (i.e., NFDS), where being the rarest phenotype had a selective advantage ranging between 1% and 20%. We ran 10,000 replicates per scenario: establishment of color polymorphism for a range of selection coefficients (0–0.2). (C) Demographic history of the kākāpō population as reconstructed with PSMC [25] and GONE [26] (Material and Methods). The vertical dashed line marks the time point of 40 generations when the 2 natural avian predators of the kākāpō went extinct. (D) Probability of the simulated color polymorphism being balanced as observed in the empirical data of the extant population under three competing scenarios: (1) NFDS is removed for all 1,000 generations, (2) NFDS is removed for the last 40 generations, or (3) same as scenario 2 plus genetic load accumulation. The dashed red line marks the probability threshold of 5% (Material and Methods). The data underlying this figure can be found in https://zenodo.org/records/13302801. NFDS, negative frequency-dependent selection.

our simulations also show that—once established—the newly evolved color morphology would have rapidly swept to fixation under pure adaptive evolution (**Fig 3A**). Hence, we also reject the hypothesis of positive selection. Instead, our simulations favor the hypothesis that balancing selection, modeled here as a trait under NDFS, is the most likely evolutionary scenario that can explain the establishment of the polymorphism in the ancestral population (**Figs 3B** and **S3C**). As the longest time for any establishment event in our simulations was 8,000 generations (mean = 5,182; SD = 500), our empirical estimate that the polymorphism has arisen 128,500 generations ago would have given plenty of time for the polymorphism to be established in the population (**S3D** and **S4** **Figs**).

Finally, we modeled the dynamics of the maintenance of the color polymorphism during the last 1,000 generations of the declining demographic history of the kākāpō, including the severe bottleneck before 1995 (**Fig 3C** and Material and Methods). The kākāpō ecosystem has changed dramatically in the past approximately 600 years (40 generations), with several indigenous bird species including the two only natural kākāpō predators going extinct within 200 years of arrival of tūpuna Māori, roughly 700 years ago [14].

Given this extinction of the kākāpō's natural predators and given that our optical analyses pointed to a potential role of predation due to differential light reflectance between green and olive feathers, we here tested under which condition the polymorphism could have been maintained in the extant kākāpō population in the absence of NFDS after being established by NFDS. We tested the following competing scenarios using computer simulations: (1) NFDS was completely removed for all 1,000 generations (as negative control); (2) NFDS was removed for the last 40 generations (modeling the extinction of the kākāpō's natural predators); or (3) NFDS was removed for the last 40 generations, and in addition genetic load accumulated in

alternative color haplotypes to mimic the effect of associative overdominance (**Fig 3D**). We were able to marginally reject scenario 1 since under pure genetic drift the polymorphism is often lost ($p < 0.05$). Scenarios 2 and 3 led to a much higher rate of balanced polymorphisms, with associative overdominance increasing the chances of this happening (**Fig 3D**). The observed color polymorphism might have therefore been maintained by chance without any form of balancing selection in the past 600 years.

## Discussion

By combining deep genomic and detailed phenotypic data with computer simulations and optical analyses, we have studied the origin and evolution of the kākāpō feather color polymorphism. We describe the putative genomic architecture of this phenotype and establish that a dominance epistatic interaction of 2 biallelic single-nucleotide genetic variants can explain the phenotype of all 168 birds, which had paired genomic and phenotypic data. Integrating this genomic architecture with genomic forward simulations, we explain the likely evolution and puzzling maintenance of this balanced polymorphism in the heavily bottlenecked kākāpō population.

Established genetic polymorphisms are rare and challenging to identify due to the need to rule out neutral evolution and determine the evolutionary forces maintaining them, such as balancing selection [11]. Rejecting neutral evolution requires assessing the genetic basis, fitness consequences, and potential selective agents of the phenotype. If directly measuring fitness effects is not feasible, balancing selection can still be inferred by rejecting neutral evolution and positive selection as explanations. Here, we estimated that the kākāpō feather color polymorphism was roughly established around 1.93 million years ago as an oligogenic trait, with green as the likely ancestral phenotype. Through genomic simulations, we found that neutral and positive evolution were highly unlikely, suggesting balancing selection through NFDS as a possible driver in establishing this polymorphism.

We hypothesize that the selective agent of such balancing selection could have been the 2 only natural kākāpō predators, the Haast's eagle and Eyles' harrier, which evolved at roughly the same time as our observed genetic polymorphism, approximately 2 million years ago [15] and went extinct approximately 600 years ago [14]. The Haast's eagle evolved in the late Pliocene/early Pleistocene when it diverged from the little eagle 2.22 MYA (95% credibility interval of 1.41–3.25 MYA), coinciding with the increase of open woodlands and grasslands at the onset of Pleistocene glaciations in Aotearoa New Zealand about 2.5 MYA. The Eyles' harrier diverged from the spotted harrier around 2.4 million years ago [15]. Our hypothesis is that the kākāpō color phenotypes were maintained by balancing selection, i.e., NFDS, through the search image formation of these avian predators who are known to mainly hunt by sight [16]. This mechanism allows predators to detect cues associated with the prey species, and in this sense being the rarer phenotypes confers an adaptive advantage. Our simulations further show that after the extinction of both predator species circa 600 years ago, the genetic polymorphism could have been maintained due to the inherent inertia in allele frequency changes. The genomic signature of the color polymorphism and the dating of its origin and evolutionary trajectory therefore provide the first evidence that the color polymorphism in the kākāpō might be a relic, or "ghost," of balancing selection by now-extinct apex predators.

We emphasize that other mechanisms of balancing selection [11], or other selective agents than predation [27], could have led to historical and/or ongoing balancing selection in the kākāpō population. For example, mutualism and parasitism are well-known mechanisms of NFDS [27]. We, however, argue that the color polymorphism as a visual target of NFDS makes predation a likely selective agent (as opposed to, e.g., host–parasite interactions), especially in

combination with the roughly coinciding evolutionary timeframe of color polymorphism establishment and evolution of the only 2 natural kākāpō predators.

NFDS due to predation might hereby stem from differences between green and olive feathers in the visible or UV spectrum. Our potential candidate gene *LHX8* and its known role in tissue differentiation and tooth morphogenesis led us to suggest a structural morphology as the source of the kākāpō color polymorphism: Structural colors are prevalent in birds and have evolved numerous independent times, and, different to simple pigmentation colors, are produced by nanometer-scale biological structures that differentially scatter, or reflect, wavelengths of light [21]. Interestingly, Dyck noted many years ago that in particular UV-modulating green and olive colors in parrots are produced by structural differences in feather barbs and barbules [28]. While we were only able to assess the photoreflectometry of 1 green and 1 olive feather, respectively, this analysis provided some evidence of differential UV reflectance between green and olive feathers, which might be explained by the relatively smoother barbule surface of green feathers according to our SEM image analysis of altogether 6 feathers. This differential UV reflectance could potentially explain how the derived olive phenotype could have become established by providing an initial evolutionary advantage through reduced UV reflectance, before balancing selection would have maintained the established color polymorphism. We, however, note that while there is some evidence for widespread UV sensitivity across bird species [22], the inference of UV sensitivity from phylogeny is complex in the avian clade [29], so that we cannot assume with certainty that the kākāpō's 2 naturally predators, the Haast's eagle and Eyles' harrier, were able to see light in the UV spectrum.

With respect to the identified candidate gene, we emphasize that no functional experimental validation of the genomic architecture or of the potential candidate gene is possible in this taonga and highly protected species. We, therefore, acknowledge that it remains a possibility that we have failed to identify the actual causal genetic variation. While the mean sequencing depth is stable across our genomic region of interest, several individual genetic sites show a sudden drop in coverage; we could therefore have potentially missed short causal genetic variants such as SNPs and indels in our analysis. As our genetic algorithm that predicts the potential dominant-epistatic interaction between our 2 indicator SNPs is computationally expensive, we further only focused on our genomic region of interest (at the end of chromosome 8). This means that we ignored the potentially modulating effects of genetic variants outside of this genomic region. Because of the reliance on short-read sequencing data, we might have further missed the impact of structural variation, which could be assessed more robustly using long-read sequencing technology in the future. When it comes to the functional annotation of our genomic region of interest, we have further found that the gene annotation of the utilized kākāpō reference genome lacks the annotation of some open reading frames [2]. An inadequate annotation at the end of chromosome 8 could therefore have prevented us from identifying the correct causal gene or gene pathway.

Our understanding of the evolution and maintenance of the kākāpō color polymorphism could potentially inform ongoing conservation efforts. Especially given that the significant majority of the wild founder population from 1995 were of green color, the color polymorphism might impact fitness in the absence of current intensive conservation management. Assuming that balancing selection has regulated the polymorphism through now-extinct apex predators, we contend that its loss would probably not confer a negative fitness effect or cost to population viability in the present-day environment. Without predator-mediated NFDS or any other form of balancing selection and without any intense conservation management, we predict that the polymorphism is likely to be lost by drift within the next 33 generations (95% confidence interval of 10 to 94 generations). In view of the aim of the kākāpō conservation management to re-establish the species on a future predator free Aotearoa New Zealand

mainland, our results might suggest that the feather polymorphism is not necessarily a trait that has to be prioritized by the conservation program for the new founding population. As this re-establishment has just taken its first step with the historic release of ten kākāpō to Sanctuary Mountain Maungatautari, Waikato (a fenced reserve on the mainland) in 2023, we are hopeful that our and others' genomic analyses might play a role in future kākāpō conservation efforts in partnership with Ngāi Tahu.

## Material and methods

### Phenotypic analyses

As part of the Department of Conservation Kākāpō Recovery Programme in partnership with Ngāi Tahu, we assessed the color polymorphisms of the extant kākāpō population. We developed a standardized protocol to be used whenever a kākāpō individual undergoes their regular health monitoring: we used a mobile phone camera (Samsung S20 FE; flash switched on) to take photographs through a short white plastic pipe while directing the focus on the feathers, ensuring that natural light was kept out to standardize luminance as much as possible. For every bird, we chose an area of undisturbed feathers on the back of its neck that contained as many feather tips as possible. We also included historical photographs of deceased birds, resulting in an extensive color catalog of 192 individuals (**S1 Table**, Photography sheet).

While most green and olive individuals could be distinguished by eye (**S1 Table**, GWAS sheet for final phenotypes), some color polymorphisms could not easily be determined. We therefore performed CIELAB color space analysis on those individuals for which we had obtained both, genomic data and standardized photography (**S1 Table**, CIELAB sheet), as implemented through an open-access application programming interface (API) provided by "Image Color Summarizer 0.76 2006–2023 MartinKrzywinski" and accessible via http://mkweb.bcgsc.ca/color-summarizer/ [last access: 13/02/2023; 23:00 CET]. The CIELAB color space analysis projects any color into a 2D color space $a*b$ that is perpendicular to luminance $L$. As $L$ is therefore an independent axis perpendicular to the color axes $a$ and $b$, differences in luminance will not have an impact on the $a*b$ projection of the photograph. The color axes $a$ and b describe the green-red and blue-yellow color components, respectively. In the case of the kākāpō feathers, a negative median of a of approximately −10 and a wide spread of a of approximately >20 defines the green phenotype (**S5A Fig**), while a very sharp $a$ peak (spread of a of <20) at a median of approximately 0 defines the olive phenotype (**S5B Fig**), with the $b$ distribution being relatively constant across both phenotypes (**S5 Fig**). We used the difference in spread of a to assign kākāpō individuals to their color, with a sharp peak (spread of a <20) close to zero (median of a >-10) defining the olive phenotype (**S1 Table**, GWAS sheet for final phenotypes).

### Genomic analyses

Our analyses are based on high-throughput paired-end Illumina sequencing libraries from nucleated blood samples for 169 kākāpō individuals at a mean coverage of 23×. Based on 2,102,449 SNPs and 417,571 indels obtained from using a deep convolutional neural network as variant caller as implemented in DeepVariant [2,17] and the high-quality kākāpō reference genome bStrHab1.2.pri (NCBI RefSeq assembly accession number GCF_004027225.2) [2,7], we filtered the genetic variant set for sequencing depth and quality using BCFtools [30] v1.9 (QUAL>20 and FMT/GQ>10 and FMT/DP>3) and for MAF (>5%) and genotype missingness (<20%). We additionally filtered the SNP set for biallelicity, for the alleles A, C, G, and T, and lenient HWE deviations ($p < 10^{-7}$). This resulted in 1,211,090 biallelic SNPs and 180,129

indels. We conducted PCA in PLINK [31] v1.9 to calculate the first 10 principal components (PCs) of the genomic covariance matrix based on the SNP set.

For the genomic region downstream of chr8_63000000, we further performed the following analyses. We calculated pairwise LD ($r^2$ between each SNP pair) using PLINK [31] v1.9. We used SAMtools [30] v1.12 to calculate the per-base sequencing depth across all samples (samtools depth). We further used BEDTools [32] v2.29.2 to average the depth across sliding windows of length 10 kb and with a step size of 1 kb (bedtools makewindows and bedtools intersect). We further accessed the bStrHab1.2.pri genome assembly (NCBI RefSeq assembly accession number GCF_004027225.2) and its associated RNA-seq exon coverage data (aggregate (filtered), Annotation release 101) to annotate and visualize gene annotation.

## Genome-phenome analyses

We used a Bayesian mixture approach implemented in BayesR [18] to estimate the heritability of the color polymorphism (BayesR version: https://github.com/syntheke/bayesR.git [last access: 13/02/23, 23:00 CET]). BayesR models SNP effects as a mixture of 4 distributions, with 1 SNP category of effect size 0, and the 3 other SNP categories reflecting distributions around SNP effect sizes of 0.0001, 0.001, and 0.01 of the overall phenotypic variance; this is important since BayesR allows for SNPs to have zero effects on the phenotype, which is often not the case for other models that assume small but existing contributions to the phenotype from every single SNP, interfering with the heritability estimation of mono- or oligogenic phenotypes. The close relatedness between kākāpō individuals does not pose a problem for BayesR since it informs the additive genetic variance that the mixture approach tries to detect. We ran the Markov chain Monte Carlo (MCMC) for 200 k cycles (after which low autocorrelation of MCMC convergence was achieved), with a burn-in of 50 k cycles, and sampled every 100th cycle. To assess polygenicity, we then conducted chromosome partitioning by partitioning the estimated heritability across all chromosomes. We also used BayesR to perform within-population phenotypic prediction using 10-fold cross-validation; we assessed the performance by one AU-ROC estimated across all 10 splits.

To prevent spurious phenotype-genotype associations, we conducted GWAS using RepeatABEL [19] v1.1 (see Data and code availability) where we modeled the relatedness matrix between individuals as a random covariate. RepeatABEL has been shown to be suitable for binary data, but its power to detect causal SNPs decreases when the proportion of successes for the analyzed trait decreases [19]. To assess its robustness, we compared the results with the standard tool PLINK [31] v1.9, which can explicitly model binary phenotypes but which only takes into account fixed effects. We therefore modeled the first 10 PCs of the genomic covariance matrix as fixed effects.

We applied a custom genetic algorithm to the genomic region surrounding the 2 indicator SNPs, namely to all 1,100 SNPs located downstream of chr8_63000000 (i.e., starting from chr8_63000328 to the last SNP on the chromosome, chr8_63426960), to explain the genomic underpinnings of the color polymorphism of all kākāpō individuals. The algorithm uses a matrix of predictor variables (i.e., the 1,100 SNPs) placed in columns, with the sample sequences in rows. An additional column that contains the binary codes for the phenotype (0 for green, 1 for olive) acts as a target value. The genetic algorithm starts by generating a set of $n$ (where $n$ is user-defined) random expressions of the form SNP001 = "C," which are then evaluated against each sequence as true or false. Each of the random expressions is then scored against the target expression using a penalized phi-coefficient. The penalty reduces the phi-coefficient by $(1 - p)$, with $p$ as the proportion of sequences in the alignment with missing data for the given nucleotide position. The lowest $n$ (user-defined) scoring rules are discarded—the

remainder are preserved as the "breeding population." The algorithm then loops through a user-defined number of "generations," typically 1,000 to 5,000, using the breeding population from which to mutate or recombine new predictive expressions, plus a number of entirely new random expressions. Expressions can be combined using Boolean operators (AND, OR, NOT, or AND NOT) as conjunctions, thereby stimulating possible non-additive interactions, including dominant epistatic interactions between multiple SNPs. These are evaluated as described above. The populate of expressions "evolves" to give the best set of predictive expressions. The only expression of Boolean Operators that could explain all color phenotypes correctly was Chr8_63055688 = T AND Chr8_63098195 = G to predict the olive phenotype. The algorithm is written in Transect-SQL and implemented in Microsoft SQL Server version 14.0.2002.14. For further detail, see Smallbone and colleagues [20].

## Out-of-sample analysis

We analyzed publicly available genomic data of a museum specimen of a yellow kākāpō (specimen ID: Av2059, host scientific name: Canterbury Museum (NZ), specimen voucher ID: 2059) [7]. We downloaded the raw single-end Illumina sequencing data from the European Nucleotide Archive (ENA; project ID: PRJEB35522; sample accession ID: SAMEA6244414), and then used BWA [33] v0.7.17 for alignment of all fastq files to the kākāpō reference genome[2], SAMtools [30] v1.12 to merge all files, picard v2.21.8 (http://broadinstitute.github.io/picard [last access: 13/02/2023, 23:00 CET]) to remove read group information, and GATK [34] v4.1.8.1 for variant calling. We subsequently used BCFtools [30] v1.9 to obtain the genotypes of our SNPs of interest.

We further received access to the genotypes of the 2 indicator SNPs in the next generation of kākāpō chicks (hatched after our data collection's deadline on January 1, 2018). This genomic data has been generated and processed according to the pipeline described by Guhlin and colleagues [2], but is not yet publicly available.

## Evolutionary origin analyses

We identified the putative ancestral allele of our SNPs of interest by leveraging the high-quality reference genome of the most closely related species, the kea (*Nestor notabilis*) [23]. We downloaded the publicly available kea raw paired-end Illumina sequencing data from the National Center for Biotechnology Information (NCBI; accession numbers SRX341179/80/81) to map the reads to the kākāpō reference genome. Raw data processing and analysis was done as described for the yellow museum specimen (see Material and Methods, Out-of-sample analysis).

We then estimated when the polymorphism might have arisen in the kākāpō based on the number of SNPs and the total number of nucleotides in the candidate genomic region, assuming a mutation rate of $1.33 \times 10^{-8}$ substitutions/site/generation (note that both SNPs are putatively located in noncoding genomic regions) [7]. We estimated a length of 2,081 nucleotides for the sum of the haplotypes carrying both indicator SNPs. While the 2 SNPs are in strong LD with each other, they are separated by SNPs that are not in LD with either of the polymorphisms (**S2A Fig**). As each indicator SNP is therefore the only SNP on its haplotype, we estimated their haplotype lengths by halving the distance to the next neighboring SNP, resulting in a length of 1,243 nucleotides for the Chr8_63055688 haplotype and a length of 839 nucleotides for the Chr8_63098195 haplotype. We then used a binomial cumulative distribution and an estimated generation time of 15 years to estimate the likely age of the polymorphism.

## Genomic simulations

We performed individual-based, forward in time simulations with a Wright–Fisher implementation in SLiM3 [24]. We simulated a genomic region of 50 k nucleotides around our 2 indicator SNPs. We scored the color phenotype of all simulated individuals based on the putative epistatic interaction of the 2 indicator SNPs, using information based on the empirical data that specifies that an individual is olive if it has at least 1 derived allele at both genetic loci.

**Demographic trajectory.** To simulate the demographic trajectory of the kākāpō population, we first reconstructed its effective population size $N_e$ trajectory during the last 1,000 generations by combining an existing PSMC [25] estimate based on the kākāpō reference genome [7] and an LD-based estimate with GONE [26] with our population genomic data set of 168 individuals (recombination rate of 2cM/Mb). While GONE can reliably estimate $N_e$ until approximately 200 generations into the past or 3 KYA, PSMC analyses are restricted to making inferences between approximately 20 KYA and 1 MYA ago. To connect the recent and ancient $N_e$ trajectories, we assumed that the estimated trajectory of steady decline of the kākāpō population continued between 20 and 3 KYA.

**Establishment simulations.** We modeled the conditions required to establish a novel olive polymorphism in a large ancestral population. The size of the simulated ancestral population of 36,000 birds was determined as the largest historical population size by PSMC [7,25], at approximately 2 MYA which was when we estimated the color polymorphism to have evolved. We assumed 1 variant (SNP1) was segregating at a low frequency of 0.4% in the population before the second SNP (SNP2) occurred as a de novo mutation. We chose the initial frequency of SNP1 (0.4%) by performing a set of simulations with neutral mutations for the same large ancestral population size and took the median MAF at equilibrium. To assess the impact of different MAFs at SNP1, we ensured that rerunning our simulations assuming a gradually increasing MAF (up to 10%) would result in similar conclusions (**S4 Fig**). We further assumed that SNP2 appeared because of a random point mutation in this genomic background. If the polymorphism was lost, we would restart the simulation with a new seed number. Otherwise, we ran each simulation for a maximum of 133,000 generations or 2 MYA. We tracked the dynamics of the polymorphism every 500 generations and if the polymorphism was either balanced (proportion of 0.4 to 0.6) or (nearly) fixed (proportion > 0.95) for 10 consecutive times (5,000 generations), we stopped the run and considered the polymorphism as established.

We tested 3 alternative hypotheses for the establishment of the color polymorphism: (1) a neutral model, where being either green or olive color had no selective advantage or disadvantage; (2) a positive selection model, where the novel olive phenotype had a selective advantage ranging between 1% and 20%; and (3) a balancing selection model, where being the rarest phenotype had a selective advantage ranging between 1% and 20% (i.e., NFDS). We ran 10,000 replicates per scenario.

**Maintenance simulations.** Starting from an established balanced polymorphism in the ancestral population (proportion of 0.4 to 0.6 of olive individuals), we tested which conditions could lead to the maintenance of such a polymorphism while considering the declining demographic history of the kākāpō population, ecological changes, and the impact of genetic drift for the last 1,000 generations. We seeded the maintenance simulations with the output of the previous step (i.e., establishment simulations) that included a large ancestral population with a balanced color polymorphism. A total of 100 starting input files were sampled at random to seed the maintenance simulations to account for a range of starting variation. Since balancing selection (NFDS) is required to establish the polymorphism, we tested whether the polymorphism could be maintained as observed in the extant kākāpō population when removing the effect of NFDS.

We simulated 3 competing scenarios: (1) NFDS was completely removed for all 1,000 generations leaving the polymorphism to drift, representing a control treatment where only drift is at play; (2) NFDS was removed for only the last 40 generations (i.e., the approximate time of predator extinction), leaving the polymorphism to drift just at the very end of the simulation; (3) NFDS was removed for the last 40 generations as before, and additionally genetic load accumulated in alternative color haplotypes to mimic the effect of associative overdominance. In this last scenario, the assumption is that the alternative color haplotypes accumulated recessive highly deleterious mutations (s = 0.2) at different loci. Selection would then act against individuals that are homozygous and expressing the deleterious effects of these mutations, and this would maintain a balanced polymorphism.

To compare the simulated and empirical data, we first calculated a metric to express the level of the balanced polymorphism (BP) as $BP = 2 - 2(G^2 + O^2)$, where G is the proportion of green individuals and O is the proportion of olive individuals. The value of BP ranges between 0 (no polymorphism, with one phenotype being lost and the other fixed) and 1 (completely balanced polymorphism with equal numbers of green and olive). The BP in the empirical data is 0.992, reflecting a value close to a perfect balanced equilibrium. Then, we compared the BP of the last step of the simulation (i.e., the extant population) to that of the empirical data. We ran 10,000 replicates per scenario.

## Functional analyses

We subjected green ($n = 3$) and olive ($n = 3$) feather tips to SEM to study their morphology in detail; we worked with uncoated samples to not destroy the culturally valuable feathers. The feathers were attached to the SEM stage using double-sided carbon tape. They were then imaged in a JEOL 6700F field emission SEM. We used an accelerating voltage of 3 kV and the lower secondary detector for imaging. This detector mixes secondary and backscatter electrons together, which lessens the charging effect for some uncoated samples. We iteratively increased our SEM resolution from 40× to 5,500×, which allowed us to focus on individual barbules. To quantitatively assess the smoothness of the feather barbules, we used Phyton's OpenCV (https://github.com/opencv/opencv; v4.9.0) function *quantify_smoothness* to calculate the average gradient magnitude: A lower gradient magnitude hereby describes higher smoothness. We first converted the SEM pictures to grayscale, chose a rectangle of the barbule of each feather at 5,500× resolution (**Fig 2A**), and then computed the gradient magnitude via the Sobel operator by taking the square root of the sum of squares of the gradients in horizontal and vertical directions.

We further subjected the feather tips of one green and olive feather to near-infrared/visible photoreflectometry. We used a dual-beam photospectrometer of the model Shimadzu UV-3101PC with MPC-3100 reflectometry accessory. The machine shines light of a single wavelength on the sample, measures how much of it is reflected, and then changes the wavelength and repeats. Briefly, feathers were clamped against the side of the integrating sphere of the spectrometer to then record the reflectance as the scanned wavelength. The measurements were standardized against a white reference (Barium Sulphate; $BaSO_4$) so that 100% reflectance means that all light is being reflected and 0% reflectance means that no light is being reflected. The beam size was a rectangle of 15 mm × 5 mm.

## Supporting information

**S1 Table. Catalog of kākāpō color polymorphisms.** GWAS sheet: Color polymorphism of all kākāpō whose genomes have been sequenced, and their genotypes at the 2 indicator SNPs associated with the color polymorphism (0 = homozygous reference; 1 = heterozygous;

2 = homozygous alternative). Photography sheet: Standardized photography of the entire extant species (as of 01/01/2018) and historical photographs of deceased individuals. CIELAB sheet: Results of the Commission on Illumination $L^*a^*b$ (CIELAB) color space analysis, which projects any color into a 2D color space $a^*$b perpendicular to luminance $L$. We annotated all kākāpō individuals for which we had any doubt in terms of their phenotypic classification. The code to generate this figure can be found in https://zenodo.org/records/13302801. For data, see the "Data and code availability" section.
(XLSX)

**S2 Table. List of all genes located on chromosome 8 between 63 and 63.4 Mbp according to the kākāpō reference genome annotation (see Material and methods and S1C Fig).**
(XLSX)

**S1 Fig. Genome and genome-phenome analysis of the color polymorphism of 168 kākāpō individuals.** (A) Global genomic PCA, colored according to the color polymorphism of the individual kākāpō. (B) Chromosome partitioning of color polymorphism heritability according to BayesR (Material and Methods); only the largest 8 chromosomes are annotated. (C) ROC and AU-ROC of within-population 10-fold cross-validation when predicting the color polymorphism from genome-wide data using BayesR; all predictions across 10 training/validation splits of approximately 17 individuals each are shown. **The code to generate this figure can be found in https://zenodo.org/records/13302801. For data, see the "Data and code availability" section.**
(TIFF)

**S2 Fig. Genomic region on the kākāpō reference genome chromosome 8, 63 to 63.4 Mbp, which contains the two indicator SNPs of the color polymorphism, Chr8_63055688 (here SNP1) and Chr8_63098195 (here SNP2). The 2 indicator SNPs are shown by vertical red lines.** (A) Heatmap of pairwise LD between all SNPs in the genomic region. (B) Sequencing depth of the genomic region (blue: per bp; brown: mean per sliding window of size 10 kbp and step size 1 kbp). (C) Gene annotation and RNA-seq exon coverage of the region (Material and Methods; S2 Table). For the data underlying this figure, see the "Data and code availability" section.
(TIFF)

**S3 Fig. Genomic forward simulation results of the establishment of the color polymorphism (i.e., no loss by genetic drift) in a large ancestral population ($N_e$ = 36,000), assuming (A, B) positive selection (left column), (C, D) balancing selection (right column), and neutral evolution (selection coefficient s = 0).** The polymorphism dynamics were assessed every 500 generations and if the polymorphism was either balanced (proportion of 0.4 to 0.6) or (nearly) fixed (proportion >0.95) for 10 consecutive times (5,000 generations), it was considered as established. (A) Percentage of simulation replicates that established the color polymorphism under neutrality (s = 0) and positive selection (s > 0). (B) Generation to establishment of the color polymorphism under neutrality (s = 0) and positive selection (s > 0). (C) Percentage of simulation replicates that established the color polymorphism under neutrality (s = 0) and balancing selection (s > 0), i.e., NFDS. (D) Generation to establishment of the color polymorphism under neutrality (s = 0) and balancing selection (s > 0), i.e., NFDS. The data underlying this figure can be found in https://zenodo.org/records/13302801.
(TIF)

**S4 Fig. Percentage of simulation replicates that established the color polymorphism (i.e., no loss by genetic drift) in a large ancestral population ($N_e$ = 36,000).** Different simulations

assume different MAFs (columns) of the first SNP segregating in the ancestral population before the second SNP occurred as a de novo mutation (under neutral evolution (selection coefficient s = 0), balancing selection (i.e., NFDS, top row), and positive selection (bottom row)). The polymorphism dynamics were assessed every 500 generations and if the polymorphism was either balanced (proportion of 0.4 to 0.6) or (nearly) fixed (proportion >0.95) for 10 consecutive times (5,000 generations), it was considered as established. The data underlying this figure can be found in https://zenodo.org/records/13302801.
(TIF)

**S5 Fig. Exemplary results of the CIELAB color space analyses of the kākāpō feathers, that project any color into a 2D color space $a^*b$ that is perpendicular to luminance $L$.** The color axes $a$ and $b$ describe the green-red and blue-yellow color components, respectively. (A) CIE-LAB analysis of a green feather, defined by a negative median of $a$ of approximately −10 and a wide spread of approximately >20. (B) CIELAB analysis of an olive feather, defined by a sharp $a$ peak (i.e., spread of <20) at a median of approximately 0. The $b$ distribution remains relatively constant across both color morphologies. We used this CIELAB analysis to assign any individual with a sharp peak of a (spread of a <20) close to zero (median of a >-10) as olive.
(TIF)

## Acknowledgments

This work has arisen from a partnership between Ngāi Tahu and the Aotearoa New Zealand Department of Conservation Te Papa Atawhai (DOC), the Kākāpō Recovery Programme, and the involved researchers. We thank the Kākāpō125+ Project led by DOC in partnership with Ngāi Tahu, the Genetic Rescue Foundation, University of Otago, New Zealand Genomics Limited, Rockefeller Institute, Duke University, Science Exchange and Experiment.com for the generation and availability of the data used in this study. Thanks for their support with technical challenges and sample/data access goes to Elizabeth Girvan from the SEM facility, University of Otago, the University of Otago Physics Department, the Dodd-Walls Centre for Photonic and Quantum Technologies, Paul Scofield from the Canterbury Museum (also for the lovely tour), and Nic Dussex from the Swedish Museum of Natural History. Thanks for their support with computational analyses goes to the New Zealand eScience Infrastructure (NeSI), especially Dinindu Senanayake, and to Mirte Bosse, René Malenfant, Lars Ronnegard, Christiaan de Leeuw, and Françoise Thibaud-Nissen. We thank the Kākāpō Recovery team: Karen Andrew, James Bohan, Nichy Brown, Jodie Crane, Galen Davitt, Andrew Digby, Daryl Eason, Liz Friend, Glen Greaves, Erica Hansen, Petrus Hedman, Bryony Hitchcock, Bronwyn Jeynes, Leigh Joyce, Sara Larcombe, Scott Latimer, Kate Lawrence, Sarah Little, Sarah Manktelow, Phil Marsh, Guy McDonald, Tommy McKerras, Servanne Kiss, Michael Mitchell, Ricki Ann Mitchell, Jake Osborne, Brodie Philp, Louise Porter, Tim Raemaekers, Jenny Rickett, Rachael Sagar, Alyssa Salton, Alisha Sherriff, Theo Thompson, Lydia Uddstrom, Lisa van Beek, Jason Van de Wetering, Maddie van de Wetering, Deidre Vercoe, Jen Waite, Richard Walle, Nia Weinzweig, Daniella Whitaker, and Maddy Whittaker Genomic data were collected as part of a previous study. Phenotypic data were collected for this study as part of the standard and ongoing management of the population by taking pictures and analyzing shed feathers, ensuring that no individual was subjected to any additional handling, treatment, or disturbance. Every step of the project was conducted in consultation with Māori iwi, in direct partnership with Ngāi Tahu, and with the Aotearoa New Zealand Department of Conservation Te Papa Atawhai and its Kākāpō Recovery Programme.

## Author Contributions

**Conceptualization:** Lara Urban, Cock van Oosterhout, Hernán E. Morales.

**Formal analysis:** Lara Urban, Cock van Oosterhout, Hernán E. Morales.

**Funding acquisition:** Lara Urban.

**Investigation:** Lara Urban, Lydia Uddstrom, Andrew Digby, Deidre Vercoe, Daryl Eason, Jodie Crane, Matthew J. Wylie, Tāne Davis, Marissa F. LeLec, Joseph Guhlin, Simon Poulton, Jon Slate, Farhan Azeem, Marvin Weyland, Cock van Oosterhout, Hernán E. Morales.

**Methodology:** Lara Urban, Jon Slate, Cock van Oosterhout, Hernán E. Morales.

**Project administration:** Lara Urban.

**Supervision:** Lara Urban, Anna W. Santure, Harald G. L. Schwefel, Cock van Oosterhout, Hernán E. Morales.

**Validation:** Lara Urban, Matthew J. Wylie, Tāne Davis, Cock van Oosterhout, Hernán E. Morales.

**Visualization:** Lara Urban, Hernán E. Morales.

**Writing – original draft:** Lara Urban, Anna W. Santure, Lydia Uddstrom, Andrew Digby, Deidre Vercoe, Daryl Eason, Matthew J. Wylie, Tāne Davis, Joseph Guhlin, Simon Poulton, Jon Slate, Alana Alexander, Patricia Fuentes-Cross, Farhan Azeem, Marvin Weyland, Harald G. L. Schwefel, Cock van Oosterhout, Hernán E. Morales.

**Writing – review & editing:** Lara Urban, Anna W. Santure, Lydia Uddstrom, Andrew Digby, Deidre Vercoe, Daryl Eason, Matthew J. Wylie, Tāne Davis, Joseph Guhlin, Simon Poulton, Jon Slate, Alana Alexander, Patricia Fuentes-Cross, Peter K. Dearden, Neil J. Gemmell, Farhan Azeem, Marvin Weyland, Harald G. L. Schwefel, Cock van Oosterhout, Hernán E. Morales.

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
