## [Editor Report · Decision Letter 0]

13 Nov 2023

Dear Dr Urban, 

Thank you for submitting your manuscript entitled "The ghost of selection past: evolution and conservation relevance of the kakapo color polymorphism" for consideration as a Research Article by PLOS Biology.

Your manuscript has now been evaluated by the PLOS Biology editorial staff, as well as by an academic editor with relevant expertise, and I'm writing to let you know that we would like to send your submission out for external peer review.

IMPORTANT: We think that your paper would be better considered as a Short Report. It's already nice and concise, so there's no need for any re-formatting, but please could you change the article type to "Short Reports" when you upload your extra metadata (see next paragraph)?

Once your full submission is complete, your paper will undergo a series of checks in preparation for peer review. After your manuscript has passed the checks it will be sent out for review. To provide the metadata for your submission, please Login to Editorial Manager (https://www.editorialmanager.com/pbiology) within two working days, i.e. by Nov 15 2023 11:59PM.

Kind regards,

Roli Roberts

Roland Roberts, PhD

Senior Editor

PLOS Biology

rroberts@plos.org

---

## [Decision Letter · Decision Letter 1]

17 Jan 2024

Dear Dr Urban,

Thank you for your patience while your manuscript "The ghost of selection past: evolution and conservation relevance of the kakapo color polymorphism" was peer-reviewed at PLOS Biology. It has now been evaluated by the PLOS Biology editors, an Academic Editor with relevant expertise, and by four independent reviewers. 

You'll see that reviewer #1 is very positive and only has textual and presentational requests; this includes toning down the claim for predator-related NFDS. Reviewer #2 is also positive, but wants you to clarify how the colour differences were quantified; overall this needs to be much more objective. Their remaining points are textual, though s/he raises a sceptical note about whether raptors are UV-sensitive. Reviewer #3 liked the study, but wonders whether it is better suited to PLOS Genetics; their only explicit suggestion is to make more use of the genomic data in exploring evolutionary mechanisms (s/he points you to a useful article). Reviewer #4 is also very positive, but urges caution regarding the evolutionary scenarios; like reviewer #2, s/he wants more objective/quantitative assessment of the SEM data, and he has a lot of very helpful suggestions for the genetics.

I discussed these comments with the Academic Editor, and they agree that given the overall positive tone, we should give you an opportunity to address the concerns raised. I note that due to an administrative error, reviewer #3 was sent the wrong invitation letter, and so was not aware that this was a Short Report manuscript; we apologise for this oversight.

In light of the reviews, which you will find at the end of this email, we would like to invite you to revise the work to thoroughly address the reviewers' reports.

Given the extent of revision needed, we cannot make a decision about publication until we have seen the revised manuscript and your response to the reviewers' comments. Your revised manuscript is likely to be sent for further evaluation by all or a subset of the reviewers.

**IMPORTANT - SUBMITTING YOUR REVISION**

*Re-submission Checklist*

*Published Peer Review*

*PLOS Data Policy*

*Blot and Gel Data Policy*

Sincerely,

Roli Roberts

Roland Roberts, PhD

Senior Editor

PLOS Biology

rroberts@plos.org

REVIEWERS' COMMENTS:

Reviewer #1:

In this paper, the authors present analyses of genomic data from nearly every individual kākāpō to estimate the genetic basis, origin, and maintenance of color polymorphism. They find a single SNP that explains the vast majority of green/olive polymorphism and an additional SNP that, when epistatic interactions are modelled with the first, can fully explain the phenotype. They further present some data on the phenotype, suggesting perhaps structural differences lead to the color differences. Finally, they use demographic modelling to test hypotheses about the origin and maintenance of the genetic polymorphism. Together, the results suggest that color is under balancing selection and the authors hypothesize that the selective force was extinct avian predators.

Overall this was a truly fun read. The system is charismatic and interesting. The genomic analysis is very well done and the results extremely clear. Although the color analysis is outside my area of expertise, it does strike me as a bit on the preliminary side. I also think, while the authors explanation of the agent of selection is plausible, it is far from the only possible explanation and there should be some incredibly careful caveats here. I've expanded on these comments below and added some additional minor edits: 

Color analysis: I do see the difference between the two exemplary plots in Supplementary Figure 4. However, I don't know much about how variable these outputs are. Was assignment to color type of unknown birds quantitative or was it done 'by eye'? If this was done in a quantitative way, please add more detail and also an estimate of assignment accuracy. 

Figure 1: Please include a legend for the dot size so that readers can understand the sample size used for each timepoint.

Figure 2 caption: I think there is a typo - "The genome-wide significant hit at the end of chromosome 8". Shouldn't this be chromosome 9? Chromosome 8 is the actual peak that is further explored.

Figure 2. The LD plot in the supplement is useful for understanding what is 'under' the peak on Chromosome 8, but I think it might be nice to have a zoomed in manhattan plot so we can see where the two SNPs are. 

Figure 3B: Is this just one feather of each color? How consistent is this signal?

The evidence for NFDS driven by an extinct predator is interesting, but not really conclusive. The authors rely on two main pieces of evidence here. First, that the predators evolved around the same time as the polymorphism, but the confidence interval is quite broad. Second, that simulations that include historical (but not contemporary) balancing selection can explain the maintenance of the polymorphism. However, it is also possible that balancing selection from a different selective agent that has persisted through present day could be maintaining the polymorphism. In other words, the simulations do not rule out, or even disfavor, ongoing balancing selection. I understand the logistical impossibility of experimental validation in this system! I just think the authors need to be much more careful to accurately express the level of certainty in their conclusions.

One interesting point that was not discussed is the trend shown in Figure 1B, the increase in olive individuals in recent generations. If the polymorphism is essentially neutral now how is this directional trend explained?

Reviewer #2:

General

I really enjoyed this manuscript. It's a very important topic that a diverse array of folks will find interesting- geneticists, ecologists, color science folks, etc. I appreciate the integrative approach and the salience to an important conservation issue. I found the genomics and evolutionary analyses sound. My primary issues lie with the color quantification, or relative lack thereof. I made suggestions for how to (relatively easily) quantify the color differences or at least represent the variation more objectively. I also made suggestions for how to soften your interpretation around the importance of UV.

Abstract

"such a neatly balanced color polymorphism"

Intro

Unless it's a style/formatting requirement of the journal, I might suggest removing the results from the end of the intro.

Methods

The CIELAB method is interesting, but your description lacks details about how you used the resulting data to bin an individual into green vs olive. They way you describe it makes it sound like you first assigned a feather as green vs olive and then obtained the axis value distributions. So what additional info does this provide? You need some better explanation here. Also, what is the y-axis of Supplementary Figure 4?

Some justification for the additional focus on chr8_63000000 to 63426960 would be nice here.

Some more details here about what, exactly, was quantified by the SEM and reflectance measurements are need. Also, more details about the reflectance measurements are needed. How were the feathers mounted? On what background? How many measurements were taken per individual? How were the measurements standardized? What were the spectrometer settings? How was the probe tip mounted and how close was it to the sample? Etc.

Results

The candidate gene analysis is the first place you mention that the color is structural. It would be good to mention this earlier.

Related to my methods comment about the SEM and reflectance data, the optical analyses section suffers from a lack of quantitative information. I'm not sure how to quantify "smoothness", but there are easy ways to quantify the reflectance data to compare, for example, whether there is a significant difference in UV reflectance between the morphs. You could also more objectively quantify where, exactly, the color difference is rather than focusing subjectively on UV reflectance (although that is very interesting). Related to that point, I would plot the average reflectance curve for each morph +/- SD or something to visualize variability.

Discussion

I think you could benefit from a brief discussion of the potential (or lack thereof) of balancing selection mechanisms other than NFDS to maintain the polymorphism.

Regarding the predation hypothesis, I would be careful about assuming that the raptor predators were UV-sensitive. The few raptor color vision studies suggest otherwise (Ödeen and Håsted 2003 MBE, Ödeen and Håsted 2013 BMC Evo Bio, Lind et al. 2013 J Exp Bio, Doyle et al. 2015 PLOS ONE, Wu et al. 2016 Sci Rep). These studies include Marsh Harrier and at least three eagles. This is also why my methods comment about more objectively quantifying the color difference would be helpful. Your predation hypothesis would still be supported by a "significant" contrast between the morph colors. I think you box yourself in a bit unnecessarily by focusing on UV.

Reviewer #3:

In this well-written manuscript, the authors explore the genetic basis of and the evolutionary processes that favor an interesting color polymorphism in the critically endangered kakapo. First, the authors use whole genome data to uncover the genomic regions that predict the color polymorphism, finding that a single region/SNPs in chromosome 8 explains the distinct phenotypes. The authors then explore the likely mechanisms that underlie the origins and maintenance of the color polymorphism through a combination of population genomic analyses, including genomic forward simulations in SLiM. They found that 1) balancing selection is the best explanation for the establishment of the color polymorphism, and 2) even with the removal of balancing selection coupled by a severe bottleneck, the polymorphism can persist for dozens of generations (through today). Overall, the study design and approaches are appropriate for testing the hypotheses, and I do not see any issues with them.

Other studies have also identified the genetic basis of color traits, as well as the role of balancing selection in maintaining color polymorphisms. What makes this study different from other studies are: 1) because of an extreme bottleneck, only several hundred individuals of the entire species exist; hence, the study samples almost every individual of the entire species, and 2) the combination of the extinction of an apex predator, the likely agent of selection, and a severe population bottleneck provide a unique opportunity to explore what happens to a stable polymorphism when balancing selection is relaxed and drift becomes a very important force. In addition, the kakapo is a critically endangered and culturally important species, and thus is a conservation priority. This study, therefore, provides a nice example of the persistence of a polymorphism after intense selection is removed in an important avian species. Overall, the work adds to the large body of literature showing the role of balancing selection, especially by predators, in maintaining stable polymorphisms in the wild. With this in mind, this study is very interesting and should be published, but I am not entirely convinced the results meet the novelty and outstanding criteria required by PLoS Biology (e.g., a strong case for publication in PLoS Genetics can be made).

Other points:

I appreciate that the authors were open about the limitations of their short-read, whole genome data set - including the inability to test for structural variants in predicting phenotype. This is an important avenue of research for the future to validate their candidate gene/genomic region.

The use of SliM to test different evolutionary scenarios is appropriate, but I would like to have seen even more effective use of whole genomic data. There are several ways to do so, which was recently detailed in a comprehensive review by Bitarello and colleagues in GBE.

Reviewer #4:

I enjoyed reading this paper about the origins of the kākāpō plumage polymorphism. The genomic analyses are nicely done and I find the QTL analysis to be highly convincing. The story regarding the predators are a fascinating dimension that are generally supported by the results, but I also urge some caution when interpreting the deep timing of these events and also feel that alternative scenarios to explain the polymorphism would be worth discussing (see my comments about the discussion). There are two results in particular that would benefit from further clarification or analysis, which is the invocation of epistasis for the two top QTls and the mechanistic basis for color variation at the feather barb level. I describe my thinking on these two points and other, relatively minor points, below. 

My copy did not include line numbers so I do my best to outline the context with my comments. 

Page 2: "improving our understanding of the evolutionary forces acting on the genomes of this threatened species" In the context of this paper, the evolutionary forces (i.e. natural selection) are acting on the phenotype, rather than the genome per se. 

Page 2, bottom. Some more background on why a polymorphism is expected to be lost would be beneficial to readers. 

Page 5, bottom: I find it unconvincing not to include the indel data analysis here, especially when an indel could be linked to one of the associated SNPs, while not passing significance itself, e.g. if indels are called with higher false negatives than SNPs as is often the case. 

Fig 2 caption: the discarded SNP is said to be on chr8, did you mean 9 (it looks blue to me)?

Page 7: "Out of sample predictions" I found this first paragraph challenging to follow. The order of which SNP is shown first changes in the initial description of Te Atapo (195, 688) to the rest of the paragraph (688, 195). Consistency would help here, while an additional figure illustrating the pedigree and genotypes I think would make it much clearer to the reader. If I read this correctly, all green individuals are TT at Chr8_63098195? Then it is unclear to me what the out of sample analysis tells us with regards to the two SNPs - but this may just be my misunderstanding of the data as it is presented. 

Fig 3, I have a hard time appreciating the described smoothness from the SEM images. Can this be quantified and tested quantitatively? It's not clear to me which feather part is described as smooth, is it the barb or the barbules? 

Page 7: I am not sure what type of SV or uncalled SNPs would be detected from the 10kb sliding window approach. This is more typically done in this scenario by looking at depth for individuals with different genotypes, rather than looking at all individuals together. This would allow one to detect, for example, if sequencing depth is higher for one genotype or another, which could indicate a duplication or delegation associated with the SNP. With the current method, I suspect a large inversion could be detected this way, but this analysis seems too coarse to confidently discard the possibility of uncalled variation. Perhaps the authors just need to be more specific about what type of variation they are looking for. 

Page 8: "if the difference in surface modality had an impact on light reflectance" As presented, this argument feels circular. The feathers are different colors, so the spectra should differ between them, and may or may not be driven by the difference in smoothness. The authors should be a bit careful about what data they use to support the prediction that the feather smoothness itself causes the color. 

Page 8: "Using principles of evolutionary genetic theory about haplotype divergence…" I realize these can be found in the methods, but there are a number of different ways to approach this and I think it would benefit the paper to include a description of this method under results. 

Page 12: Regarding the poor annotation, given the noisy RNAseq data in Fig S1, it does seem as though the annotation is likely incomplete. If the authors are concerned that there are mis-annotated or missing genes, one option would be to use a tool such as "LiftOff" (https://doi.org/10.1093/bioinformatics/btaa1016) to liftover an annotation from a model species, even chicken. This is fast to run and would quickly reveal if any major genes are missing or of the gene name is annotated differently in a model species. 

Page 12: With regard to the conservation implications, is there a possibility that conservation intervention has somehow favored the olive phenotype? Another possibility is that the polymorphism is involved in an as-yet unmeasured behavior, such as in mate choice. This kind of interpretation, i.e. that the morphs can be ignored for conservation, is best left to the authors' discretion, but as an external reader I feel that this paragraph could be a little more careful in wording in regards to uncertainty. 

Page 13: "the four canonical bases as alleles" Is a bit of a strange filter description, are there biallelic SNPs that dont have the four canonical bases? 

Page 14: "We applied a custom genetic algorithm to all" it would be appropriate to describe here what the algorithm's intended use is - its not clear what results this paragraph corresponds to, but I believe it is used for detecting epistasis between the two top SNPs. At a minimum, a brief description of Smallbone et al 2021 seems appropriate, especially with regards to how it supports the observation of epistasis here. 

Page 15: Regarding LD and epistasis, I am not convinced by what is shown that LD is reduced between the two SNPs. My copy of Figure S1 is heavily pixelated, but the patterns of LD make it seem possible that this is a complicated genomic interval that may not be assembled correctly. Have the authors checked for assembly error that could lead to the appearance of high LD between the two SNPs but low LD between them? If there is a single pacbio contig that spans the two SNPs, than this would be an important element to add to the presentation in Fig S1 and remove this as a possible explanation. Otherwise it is difficult to evaluate conclusively if the LD between the two SNPs is epistasis, or simply linkage between two very close SNPs. The SNPs are quite close and a more thorough quantification of haplotype variation is needed to invoke epistasis. 

I wonder if the authors have checked to see if any of the called indels (e.g. small SVs found by deepvariant) are in high LD to one of the two SNPs? If the causative variant is an indel with linkage to one or both SNPs, this could also explain the pattern shown by the two SNPs. 

Page 17: SEM images and reflectance. It looks as though feather reflectance was measured on multiple feathers. A key argument for the role of predators is that UV reflectance differs between olive and green feathers. It would seem appropriate to run a statistical test utilizing reflectance in the UV of multiple feathers to make the argument that reflectance is consistently different in this range, and not a outcome of measurement error. Reflectance values can differ considerably depending on the placement of probes on feathers, surface background, and the angle of the feather- using repeated measurements and multiple feathers is one way to reduce this potential for error. 

Page 11: Discussion of UV results: given that the feathers reflect light in both visible and UV, its not clear to me the significance of the difference in the UV range. I.e. if reflectance differed only in non-UV wavelengths, would the predators still be able to observe the difference in plumage? I think the authors could expand on their argument of why UV reflectance per-se points to predators as the causal selective agent.

---

## [Editor Report · Decision Letter 2]

5 Jul 2024

Dear Dr Morales,

Thank you for your patience while we considered your revised manuscript "The ghost of selection past: evolution and conservation relevance of the kakapo color polymorphism" for publication as a Short Report at PLOS Biology. This revised version of your manuscript has been evaluated by the PLOS Biology editors and the Academic Editor.

Based on our Academic Editor's assessment of your revision, we are likely to accept this manuscript for publication, provided you satisfactorily address the following data and other policy-related requests.

IMPORTANT - please attend to the following:

a) Please change your Title to "The genetic basis of the kākāpō structural color polymorphism suggests balancing selection by an extinct apex predator"

b) The Academic Editor said "The only comment I have is that the new Figure 1, the Manhattan plot is difficult to read. Especially the zoomed-in insert. As much as I love the bird images, they may need to shrink them a bit to make room for the plots." I wonder if we could possibly "have our cake and eat it," by moving Fig 1B underneath Fig 1A, so that the Manhattan plot in Fig 1B is stretched to the full width of the Fig, but retaining the magnificent bird images in Fig 1A.

c) You currently say that you do not require an ethics statement. My understanding is that this is probably because the genotype data are from a previous pop gen study (ref 7), and the phenotype data are from museum samples, archival photos and routine health-checks on birds. Please could you confirm that this is the case?

d) Please address my Data Policy requests below; specifically, we need you to supply the numerical values underlying Figs 1B, 2B, 3ABCD, S1ABC, S3AB, S4, S5AB, either as a supplementary data file or as a permanent DOI’d deposition. I have previously had a conversation with Dr Urban about the restrictions on access to the kākāpō population genomic dataset and genetic variant callset, and these fall clearly under the "third party data" exemption to our data availability policy. So please provide as much of the aforementioned underlying numerical values as you can without compromising that arrangement.

e) Please cite the location of the data clearly in all relevant main and supplementary Figure legends, e.g. “The data underlying this Figure can be found in S1 Data” or “The data underlying this Figure can be found in https://zenodo.org/records/XXXXXXXX

f) Many thanks for providing your code in GitHub (https://github.com/LaraUrban42/Kakapo_genomics). However, because Github depositions can be readily changed or deleted, please make a permanent DOI’d copy (e.g. in Zenodo) and provide this URL.

We expect to receive your revised manuscript within two weeks. 

*Published Peer Review History*

*Press*

Sincerely,

Roli Roberts

Roland Roberts, PhD

Senior Editor

rroberts@plos.org

PLOS Biology

DATA POLICY:

Regardless of the method selected, please ensure that you provide the individual numerical values that underlie the summary data displayed in the following figure panels as they are essential for readers to assess your analysis and to reproduce it: Figs 1B, 2B, 3ABCD, S1ABC, S3AB, S4, S5AB. NOTE: the numerical data provided should include all replicates AND the way in which the plotted mean and errors were derived (it should not present only the mean/average values).

CODE POLICY

DATA NOT SHOWN?

---

## [Editor Report · Decision Letter 3]

16 Jul 2024

Dear Hernan,

Thank you for the submission of your revised Short Reports "The genetic basis of the kākāpō structural color polymorphism suggests balancing selection by an extinct apex predator" for publication in PLOS Biology. On behalf of my colleagues and the Academic Editor, Gail Patricelli, I'm pleased to say that we can in principle accept your manuscript for publication, provided you address any remaining formatting and reporting issues. These will be detailed in an email you should receive within 2-3 business days from our colleagues in the journal operations team; no action is required from you until then. Please note that we will not be able to formally accept your manuscript and schedule it for publication until you have completed any requested changes.

IMPORTANT: I have left the following note for my colleagues in the Production department: "Dr Lara Urban should be sole corresponding author in the final published version, but she is currently on maternity leave, so Dr Hernan Morales has been temporarily assigned as corresponding author." This should work, but please check that Dr Urban is correctly identified as corresponding author when you receive the final proofs!

Sincerely, 

Roli

Senior Editor

PLOS Biology

rroberts@plos.org